# Decoding Journalism in the Digital Age: Self-Representation, News Quality, and Collaboration in Portuguese Newsrooms

**João Canavilhas * and Branco Di Fátima ***

LabCom, University of Beira Interior (UBI), 6201-001 Covilhã, Portugal
* Correspondence: jcanavilhas@gmail.com (J.C.); brancodifatima@gmail.com (B.D.F.)

**Abstract:** This paper analyses the self-representations of Portuguese media professionals and their work practices. Utilizing data from a broader empirical study, this paper delves into the dynamics of influence among various actors within newsrooms. Based on journalists' perceptions of the content, the methods they use to assess the quality of the news are also identified. To address these enquiries, a survey was conducted among professionals engaged in the news production process. This sample comprised 72 individuals from various sectors of newsrooms, including photographers, designers, IT professionals, social media managers, and videographers. The main results indicate that seven out of ten respondents acknowledged their reliance on colleagues in newsrooms for success. Furthermore, the data suggest that there are no significant disparities among different professionals, with personal satisfaction emerging as the primary criterion for assessing the work quality. It is notable that almost twice as many women tend to indicate the low impact of the journalist on their work compared to male respondents. Moreover, most respondents stated that there is space for hybrid professionals in newsrooms.

**Keywords:** journalism; self-representation; newsroom; media professionals; news quality; convergence; new media; sociology of work; work practices; Portugal

## 1. Introduction

The rapid popularization of the internet and smartphones has profoundly altered the media ecosystem. A similar phenomenon had already occurred with the emergence of radio and television, giving rise to what is known as *mediamorphosis* (Fidler 1997) or *remediation* (Levinson 1997; Bolter and Grusin 1999). However, this time, the changes have been far more profound, and the ecosystem continues to evolve in this process of the digitalization of society.

The variation in the intensity of this transformation can be attributed to the influence of digitalization across all the phases of the news process. From information gathering (Steiner 2014), content production (Graefe 2016), and distribution and consumption modalities (Wheatley and Ferrer-Conill 2020) to the engagement of the interconnected audience, every aspect has transformed, including the journalistic profession itself (Zamith and Westlund 2022).

For years, humans have used intermediary devices to produce and exchange news. Now, automated news production by non-human actors, based on algorithms, is becoming very relevant (Zamith and Westlund 2022). By exploiting social platforms or technologies for personal use, citizens communicate and exchange symbolic content, maintaining a continuous flow of information (Castells 2009). In newspapers, for example, the advent of the Gutenberg Press and the development of transport allowed them to play a significant role in society as hubs for disseminating information (Thompson 1998). Thanks to this, newspapers and journalism have attained enormous centrality in modern societies because of their contributions for regulating the functioning of democracies (Vos and Thomas 2023).

The deep historical connection between the media and technology has facilitated the production and dissemination of news (Di Fátima 2023). Consequently, these technological changes have also influenced journalistic practices within newsrooms and the profiles of their professionals (García and Vázquez 2016). Delving into the realm of journalists specifically, it becomes apparent that the advent of new technologies necessitates professional adjustments, encompassing the adoption of methods and tools and the acquisition of technical competencies (López-García et al. 2017).

The evolving landscape demands that the media undergo processes of technological convergence (multiplatform), business convergence (concentration), and content convergence (news genres), alongside professional convergence (multi-skilling), significantly impacting journalists (Salaverría 2010). The impact of technological development on the media industry can also be understood through the dimensions of *post convergence*, including hyper-mediation, bio-digitality, hyper-connection, and hyper-simulation (Ramírez 2020).

This study delves into the phenomenon of professional convergence within journalism, particularly focusing on the practical aspects within newsrooms. In modern times, journalists frequently exhibit a hybrid profile, incorporating duties traditionally associated with other professions (Hallin et al. 2023). Such responsibilities may involve video editing, social media management, or graphic animation, all facilitated by the pervasive availability of digital technologies. Thus, hybrid professionals are those whose profile allows them to combine different skills to produce news content.

In the early stages of newspaper production, journalism and typography were the sole professions involved in the process (Azevedo 2009). Over the years, the team expanded because of the necessity for widening distribution, initially managed by the media themselves but soon becoming an autonomous sector. On the contrary, illustrators, initially external to newspapers, gradually integrated into editorial teams.

With the introduction of photography to newspapers, many illustrators were replaced by photographers. These professionals later joined newsrooms, acquiring the status of journalists—now referred to as photojournalists (Souza 2010)—a transition not experienced by illustrators. However, the significance of photojournalists in newsrooms has diminished as traditional journalists have assumed some of their roles, facilitated by digital photography.

The digitalization era and the rise of desktop publishing software led to the end of typographers, and, subsequently, rotary presses contributed to the obsolescence of these professionals in newspaper structures. Concurrently, newsrooms began to incorporate technicians with expertise in computer technology, initially tasked with addressing hardware issues but later involved in software development. It was with the advent of the internet that IT specialists and designers truly became integral to newsrooms, joining production teams regularly (Edwards 2016).

Initially, the work of these professionals was conducted in various physical spaces, each integrated into their respective sectors. Over time, the convergent newsroom model has become predominant (Meier and Torres 2010), with professionals from different sectors sharing space. In this environment of convergence, the distinctions between the cultures of journalism, technology, and the arts have become more apparent, particularly concerning the epistemological foundations of each discipline (Royal 2012).

Objectivity, a fundamental journalistic principle, holds little significance for designers—who concentrate their attention on the aesthetics of pages and graphic content—or for computer engineers—who prioritize the performance of hardware and software. The disparity between cultures is aptly described by Bell (2015, p. 35), referring to the industries in which journalists (media) and engineers (new platforms) operate: "News companies make it hard to publish; social media platforms make it easy to publish".

In the realm of design, their influence was already widespread in page layouts and graphic production. However, their significance has grown substantially alongside the media's investment in the Web. The development of tools for generating interactive content, coupled with the rise of a visually oriented culture, has propelled designers to assume

new responsibilities, including editorial autonomy. Furthermore, the rise of social media platforms and the imperative for media outlets to engage with their audience have spurred the emergence of a professional category within newsrooms: social media analysts or managers (Bergström and Belfrage 2018). This addition completes the group of techno-actors, including designers, videographers, infographic creators, and sound designers (Canavilhas et al. 2015). These professionals form modern newsrooms, ensuring their adaptability and relevance in today's media landscape, characterized by convergence culture (Menke et al. 2018).

In the past 30 years, newsrooms have undergone significant transformations, evolving into professional environments characterized by a diverse mix of cultures and perspectives (Finberg and Klinger 2014). The emergence of new professional roles aims to bridge the gap between technology and journalism (Hepp and Loosen 2021), yet this evolution has not been without tension at professional frontiers (Carlson 2015). Historically, tensions have often arisen between the established figures in newsrooms and each newcomer (Souza 2010; Godinho 2009). This phenomenon, already mapped in the literature, underscores the necessity for ongoing research to address the challenge for integrating actors and fostering relationships between professionals in newsrooms.

As outlined extensively in the Methods and Data section, Portuguese media professionals were contacted and invited to participate in this online survey. The initial set of questions aimed at identifying participants by their professional roles, offering choices, such as journalist, photographer, designer, and social media manager. Although the response rate from journalists met expectations, the same cannot initially be said for the other professional categories.

The distribution of a new email soliciting participation ignited a discourse surrounding the frontiers of journalism. The elicited reactions can be encapsulated in a statement crafted by a photojournalist from *Público*—one of the Portuguese prominent daily newspapers: "I began drafting a response but halted midway because of the survey's misguided assumption that photographers are not considered as being journalists". Similar sentiments were echoed by several designers by email. Hence, there arises a crucial need to elucidate the criteria for obtaining a journalist's accreditation card in Portugal.

Article 1 of the *Journalist's Statute* defines journalists as "individuals who, as their primary, permanent, and compensated occupation, engage in editorial functions, such as researching, collecting, selecting, and processing facts, news, or opinions, utilizing text, images, or sound for dissemination via newspapers, news agencies, radio, television, or any other electronic media" (CCPJ 2008). To obtain the mandatory accreditation card, aspiring journalists must meet three conditions: (i) being at least 18 years old, (ii) possessing full civil rights, and (iii) completing an internship at a media outlet. The duration of this internship varies, lasting 12 months for graduates in communication sciences and 18 months for graduates in other fields.

Unlike legislation in other countries, in Portugal, it is sufficient to complete compulsory schooling and fulfill the conditions outlined in the code to become a journalist. This is why Portuguese newsrooms boast professionals from diverse fields, all capable for obtaining the journalist's accreditation card, enabling them to independently sign the news pieces they produce (Rebelo 2011).

Discussions pertaining to the frontiers of journalism are not novel (Carlson and Lewis 2015). However, the advent of new media has amplified their frequency, primarily attributable to the escalating influence of technology in both the production and dissemination of journalistic content (Riedl 2023). The technological amalgamation that has immersed the media into the realm of perpetual remediation has necessitated the fortification of newsrooms with fresh expertise (Bolter and Grusin 1999). This realization emerged as professional convergence reached its high point, rendering the assignment of new roles to journalists unfeasible (Canavilhas et al. 2015).

Beyond traditional journalism, the evolving professional and technological demands within newsrooms have necessitated the involvement of individuals with diverse back-

grounds. This shift has led to the development of working models characterized by heightened interprofessional reliance (Lenzi 2017). As the digital age progresses, it is becoming increasingly crucial to conduct research on the self-representation of these emerging actors, as well as their perceptions of news quality and the complexities of collaboration within newsrooms.

## 2. Methods and Data

This study analyses the work of Portuguese media professionals based on their self-representations. Drawing upon a broader empirical study, this paper explores the relationship of influence among various actors in newsrooms, as well as the perception of the quality of news content. In pursuit of these objectives, four research questions have been formulated:

- RQ1: Does the personal satisfaction of a journalist depend on other professionals?
- RQ2: Who are the primary co-authors influencing the technical decisions?
- RQ3: Does the field of education influence how professionals assess the work?
- RQ4: Is there space for hybrid professionals in Portuguese newsrooms?

To answer these research questions, an online survey was conducted using *Google Forms*. The formulation of the questions drew from prior research (Canavilhas et al. 2015; Di Fátima 2021) that had surveyed journalists (Miranda 2018; Crespo et al. 2017). This approach guaranteed the scientific reproducibility of the study, enabling its examination and application in similar research (Morse 2010).

The research questions were designed to study the relationship between media professionals and their capacities for interdependence within newsrooms—a current gap in journalism studies. Thus, the online survey comprised 31 questions organized into four thematic blocks: (i) sociodemographic, (ii) professional satisfaction with their work, (iii) perception of the quality of the news content, and (iv) influence among professionals (journalists, techno-actors, and photographers) in newsrooms. Utilizing primary data, the online survey enhanced understanding within a realm that has received relatively scant attention in journalism studies. Employing this adapted observation tool in alignment with the outlined objectives enabled meticulous control over the empirical mechanisms of this study.

Adopting this quantitative approach, the survey link was distributed via email to 108 national media professionals, including journalists, photographers, designers, IT specialists, and social media managers. Email addresses were collected from the websites of media outlets or were suggested by editors in response to the authors. This sample included individuals from prominent daily newspapers, such as *Jornal de Notícias*, *Público*, *I*, and *Diário de Notícias*; news websites, like Observador and Semanário; magazines, such as Visão and Sábado; as well as weekly newspapers, like *Expresso*, *Novo*, and *Sol*.

To minimize the response time, a pivotal aspect in the preparation of online surveys, the questionnaire was organized with closed or semi-open questions, providing simple, scaled, and multiple-choice options (Bordens and Abbott 2011). Data collection occurred between 4 July and 21 September 2022, spanning approximately 80 days, during which the survey remained available online.

To address instances of "non-response", individuals who initially did not respond were contacted three additional times via email. This approach increased the number of responses. The survey achieved a complete response rate of 70.6% (n = 72), which stands as being notably high, especially considering the absence of financial incentives typically used to boost response rates in online surveys (Dillman 2007).

The online data collection approach facilitated outreach to individuals across various media outlets, representing a broad spectrum of the country's newsroom workforce. For some sectors of newsrooms, such as photography, it is possible to say that most professionals from the largest national news outlet responded to the survey. Thus, this sample effectively serves to illuminate empirical insights in a relatively unexplored domain (Bryman 2012; Bordens and Abbott 2011).

The outlined findings are derived from a non-probabilistic sample of media professionals, encompassing individuals of all genders in Portugal and with internet access. In terms of sociodemographic profiles, these results are consistent with findings from other national surveys focusing on professionals in newsrooms, particularly journalists (Miranda 2018; Crespo et al. 2017). The data were processed and analyzed using SPSS, v. 27.0.

This sample demonstrates gender balance, with 52.1% identifying as men and 47.9% as women. Regarding the age distribution, the participants predominantly belong to younger cohorts with some years of journalism experience. Consequently, the majority falls within the age range from 36 to 45 years old (31.0%), followed by those aged from 26 to 35 (23.9%) and from 46 to 55 (23.9%). Extremes in age representation are less significant: Individuals under 25 years old account for 14.1%, while those over 56 comprise 7.0%.

Individuals from diverse backgrounds participated in the online survey. The most prevalent group consisted of journalists (56.3%), followed by a cohort of techno-actors (28.2%)—designers, videographers, infographics creators, sound designers, etc.—photographers (8.5%), and social media managers (4.2%). For the level of education, the collected data are in line with data collected in previous studies (Miranda 2018; Crespo et al. 2017). This sample is predominantly made up of individuals with a degree (74.6%), followed by those with a master's degree (14.1%) and individuals with secondary education or equivalent (9.9%) or a Ph.D. (1.4%).

## 3. Results

The first question in the survey concerns the evaluation of professional satisfaction and the interrelation among social actors within newsrooms. The findings indicate a robust interrelation among all the professionals, with a majority asserting that their satisfaction hinges upon collaboration within newsrooms. When categorizing responses into three scalar levels (low: 1, 2, 3, and 4; medium: 5, 6, and 7; high: 8, 9, and 10), it becomes evident that the bulk of the responses gravitate toward the middle range. However, combining the medium and high levels yields compelling statistics, underscoring the extent of interprofessional dedication: 79.3% of the journalists, 73.9% of the photojournalists, 82.3% of the designers, and 75.0% of the social media managers affirm that their satisfaction is intertwined with their colleagues' contributions within newsrooms.

The respondents' gender appears to significantly influence their responses, particularly at the higher levels of the perception scale. When queried about the journalist's impact on the satisfaction of their work, nearly 60.0% of the male actors attributed moderate importance. Conversely, half of the female professionals strongly correlated the satisfaction of their work with the presence of a journalist—a scientific topic warranting exploration in future studies on the role of women in newsrooms. Moreover, it is notable that almost twice as many women tend to indicate the low impact of the journalist on their work's satisfaction compared to male respondents (30.0% vs. 15.8%). Consequently, the data suggest that female respondents more frequently gravitate toward the extremes of the scale in their responses.

The primary activity carried out in newsrooms appears to exert a more significant influence on responses when comparing various participants. For instance, the surveyed photojournalists exhibit a balanced stance regarding the importance attributed to a journalist's presence in the satisfaction of their work (33.3%). Conversely, techno-actors seem to rely more on journalists for their work, with nearly 60.0% attributing moderate importance to these media professionals. These findings could be elucidated by the dynamics of news content production within media outlets: Historically, photojournalists have enjoyed a closer working autonomy with journalists.

The second question in the survey queried which professionals (techno-actors or photojournalists) wield the greatest influence on the decision-making of journalists. Although teamwork constitutes a fundamental aspect within newsrooms, participants' answers appear to capture only a fraction of this collaborative dynamic. Among professionals exerting the most significant influence on newswriting, multimedia designers/editors emerge as

being pivotal, garnering 23.5% of the responses (see Table 1). This finding suggests a connection with the production of intricate journalistic formats, such as long-form reports, where design elements and the seamless integration of multimedia content (including photos, infographics, and videos) carry substantial weight in project success.

**Table 1.** Professionals who influence how news is made (n = 51).

| Media Professionals | Percentage (%) |
| --- | --- |
| IT Specialist | 2.0 |
| Social Media Manager | 5.9 |
| Photographer | 5.9 |
| Designer | 7.8 |
| Videographer | 7.8 |
| Infographic Specialist | 7.8 |
| Motion Designer | 7.8 |
| Sound Designer | 7.8 |
| Multimedia Designer | 23.5 |
| Other Professionals | 23.5 |
| **Total** | 100.0 |

Such emphasis may elucidate the preference for roles like sound designers, infographics specialists, and motion designers, each receiving 7.8% of the preferences, at the expense of photographers or social media managers, both tallying 5.9%. Notably, IT professionals rank the lowest, at 2.0%, often perceived merely as being technical contributors within the creative process. Further insight into this phenomenon can be gleaned by cross-referencing with sociodemographic data, revealing nuances across different categories.

In this case, gender does not significantly affect the respondents' answers. Instead, it equally underscores the importance that this sample places on multimedia designers/editors, with 18.9% male and 14.7% female. Also, the respondents' levels of education do not appear to influence their answers.

The third question of the survey pertains to how each professional assesses the quality of their work. Given the variances among professional cultures, it is crucial to ascertain whether collaborative efforts lead to a shared sense of satisfaction or if group work tends to obscure these distinctions. Because an online survey was conducted, the data reflect the perceptions of these professionals regarding their work, and generalizing these results should be approached with caution.

Self-representation appears to wield significant influence within a framework of diverse responses, grounded in the subjectivity inherent in outcome evaluation mechanisms. Personal satisfaction emerges as the primary yardstick for 28.7% of the participants in assessing the "quality of their work". Approval from external sources—be it the media outlet's editor (15.4%) or readership (10.8%)—comprises nearly a third of the responses. Despite 20.5% favoring the metric of readership, less frequently opted for by this sample are more objective evaluative tools, such as shares (10.3%) or comments (3.6%) garnered for content.

These data can elucidate why only a restricted number of interviewees emphasize the impact of social media platform metrics in assessing the quality of the news product: 35.0% of the techno-actors, 22.5% of the journalists, and 16.7% of the photojournalists. However, this reasoning is somewhat nuanced by the predominant activities undertaken by the respondents within newsrooms. In this instance, sociodemographic data assist in elucidating this phenomenon.

The activities undertaken by the respondents in the media appear to influence the tools employed to evaluate the quality of their product (multiple answers). The most notable

instance is observed among journalists (87.5%), closely trailed by photojournalists (83.3%). In other professional groups, such as techno-actors, at least 65.0% of the responses indicate "personal satisfaction". Except for photojournalists, half the respondents highlight the visibility of their work and the quantity of readers as being significant factors.

In the entire sample, data reveal that respondents under 25 years old are more likely to consider praise from the editor and media director as being the main measure of their work's quality, with 50.0% agreeing. They are also notably influenced by readers' commendations compared to other variables (19.0%). Similarly, other age groups within this sample, including those aged from 26 to 35, from 46 to 55, and over 56, prioritize praise from the editor and director. Among respondents aged between 36 and 45, however, the predominant choice is the number of comments (57.1%), albeit followed closely by praise from the editor and media director.

The two generational extremes give the least weight to the impact of social media in terms of the quality of their work: under 25 years old (9.5%) and over 56 years old (4.8%). This occurrence can be partially elucidated by the assimilation of these platforms. On one hand, the older demographic was brought up and initiated their careers in journalism prior to the Web 2.0 era, resulting in a lesser familiarity with digital technologies. Conversely, the younger cohort, having grown up in the digital age, exhibits a more discerning professional stance toward platformization.

For most of the variables, cross-referencing data by gender unveils no notable disparities in how respondents assess the quality of their work. Generally, there is minimal divergence to establish a consensus, evident in factors like the influence of social networks (0.3%), readership count (0.9%), and commendation from superiors (3.6%). Consequently, one could posit the existence of a collective perception regarding the quality of journalistic content. This shared perception persists, even in the variable displaying the most significant gender difference: personal satisfaction.

For 89.2% of the male respondents, self-representation emerges as a significant gauge for evaluating the quality of their work, whereas 67.6% of the female respondents opted for this criterion. Despite being the predominant response, signifying a shared perception, the gap between the genders stands at 21.6 points. This finding underscores the magnitude of gender disparities within the professional sphere.

The data suggest that gender affects how journalists and media professionals see themselves. This difference between men and women in Portuguese newsrooms has various reasons. Some hypotheses include fewer women in leadership roles, even though they make up the majority in some cases. Also, women face more job insecurity, especially during maternity leave. Additionally, they tend to leave the profession earlier because of unequal pay or juggling newsroom demands with household responsibilities.

The fourth and final survey question addressed the potential for hybrid professionals in newsrooms (see Table 2). These individuals would possess a diverse skill set, including journalism, photography, IT, design, and social media. Overall, 64.8% of the respondents indicated the existence of a niche for hybrid professionals, while only 7.0% chose not to respond. Additionally, 28.2% stated that they believe there is no space for this type of professional. Thus, convergence culture has expanded journalists' responsibilities to include tasks that were previously handled by other professionals in newsrooms.

**Table 2.** Space for hybrid professionals in newsrooms.

| Media Professionals | Percentage of Yes Responses (%) |
|---|---|
| Techno-actors | 85.0 |
| Photojournalists | 66.7 |
| Journalists | 65.7 |
| Average | 64.8 |

Individual samples categorized by sociodemographic characteristics reveal insightful points about the frontiers of journalism. Among these, techno-actors (85.0%) are the most inclined to believe in the presence of space for hybrid professionals in newsrooms, followed by photojournalists (66.7%) and journalists (65.7%). This result seems to affirm the notion that many techno-actors also perceive themselves as being journalists and deem their work to be journalistic, despite lacking the accreditation card to practice the profession. Epistemological questions of a similar nature had already emerged during the online survey's application. Some designers and photojournalists questioned the study team, seeking clarification on whether they had been mistakenly classified as "non-journalists".

Respondents who identified as men (73.5%) were more inclined to support the inclusion of hybrid professionals in newsrooms compared to women (65.6%). This result may reflect the greater precariousness experienced by women in the market, characterized by the insecurity of job retention. Interestingly, age appears to influence this inclination, with a higher percentage of positive responses observed among older respondents. Although 60.0% of those aged under 25 years believe in the inclusion of hybrid professionals in newsrooms, 100.0% of those aged over 56 years tend to agree with the question.

## 4. Discussion and Conclusions

The internet has expanded the horizons of journalism's narrative potential, necessitating media professionals to acquire new skills. Initially, journalists and photojournalists, professionals with a longstanding collaborative history (Garcia 2015), adeptly rose to these challenges. However, with technological advancements in interactivities and multimedia, the inclusion of additional specialists became imperative. Consequently, the demand for designers, videographers, and sound designers surged. Furthermore, the advent of new rivals stemming from Web 2.0 compelled traditional media outlets to bolster their visibility. This led to the recruitment of social media managers and IT specialists into the fold of newsrooms.

Teams, once comprised solely of two types of professionals, have evolved into multidisciplinary groups to address emerging challenges (Carvalho and Pimenta 2017). Techno-actors have introduced novel professional cultures to newsrooms, thereby necessitating an understanding of their interrelations (Canavilhas et al. 2015). Accordingly, this study employed an online survey to collect data for analyzing the interactions among various techno-actors in collaborative work environments.

First, the aim was to ascertain the degree to which each professional considers their personal satisfaction to be reliant on colleagues in other areas. Overall, other European studies indicate that journalists are satisfied with their profession because they have chosen it as a vocation (APM 2022). However, in this instance, the objective was to scrutinize labor relations within Portuguese newsrooms. At least seven out of ten participants from all the specialties acknowledge that their satisfaction is contingent upon others, with the designers' group being particularly notable (82.3%). This finding corroborates the assertion that teamwork within newsrooms is pivotal (Lenzi 2017) and is acknowledged by the participants.

The online survey also aimed to determine whether various professional cultures influence how individuals self-assess their work in newsrooms. The findings indicate that there are no significant differences among professional groups, with satisfaction with one's work being the most cited reason across this sample (n = 72). This suggests that, despite working collaboratively, the epistemological values associated with their educational background remain unchanged (Royal 2012). More important than the result or its impact on the public is ensuring that the developed piece satisfies its author.

This study also aimed to determine the professionals exerting the most influence on journalists' decisions in their professional undertakings. Among various groups, designers emerge as being particularly influential, acknowledged by journalists as significantly impacting their work. This finding aligns with expectations, considering that designers

possess technical expertise distinct from that typically held by journalists, particularly in areas such as image capturing and editing.

Finally, these results revealed that most respondents recognize the potential for integrating hybrid professionals into Portuguese newsrooms (64.8%)—even if this does not happen in practice. In this case, multidisciplinary teams seem to have gained a more obvious practical place, especially in the production of long-form multimedia content (Di Fátima 2021). Conversely, only one-third of this sample expressed skepticism about this professional profile, indicating a more conservative perception of the diversity within journalistic work. Techno-actors are the most inclined to believe in hybrid professionals (85.0%), perhaps because they perceive themselves as fitting into this social category—even if in a somewhat mythical manner.

In general, journalists see themselves as being a part of an independent group with shared goals: creating news. This perspective, emphasizing the group over individuals, defines journalism as a profession. Convergence culture is changing newsrooms in Europe and affecting Portuguese journalists (Menke et al. 2018). Journalists now handle tasks previously performed by other professionals. However, data indicate that technologically adept individuals are more open to hybrid roles in journalism.

**Author Contributions:** Conceptualization, J.C.; methodology, J.C. and B.D.F.; software, B.D.F.; formal analysis, J.C. and B.D.F.; investigation, J.C. and B.D.F.; data curation, J.C. and B.D.F.; writing—original draft preparation, J.C. and B.D.F.; writing—review and editing, J.C. and B.D.F.; visualization, B.D.F. All authors have read and agreed to the published version of the manuscript.

**Funding:** This research is funded by FCT: https://doi.org/10.54499/UIDB/00661/2020.

**Institutional Review Board Statement:** Due to the nature of the study, where an anonymous online survey was used, and the absence of personal data utilization, in accordance with the laws of Portugal, the study was deemed exempt from Ethics Committee approval at LabCom—Communication and Arts research centre, at the University of Beira Interior (UBI).

**Informed Consent Statement:** Informed consent was waived because the anonymous online survey used did not involve any personal data. However, in cases where excerpts from journalists' speeches were quoted, the research team obtained their consent.

**Data Availability Statement:** The data presented in this study are available upon request from the corresponding author.

**Conflicts of Interest:** The authors declare no conflicts of interest.

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
