# Peer review of "Decoding Journalism in the Digital Age: Self-Representation, News Quality, and Collaboration in Portuguese Newsrooms"

_journalmedia, doi:10.3390/journalmedia5020034_

Round 1
Reviewer 1 Report
Comments and Suggestions for Authors
This article analyzes the self-representations of Portuguese media professionals and their work practices. To do this, it is based on a survey conducted with 72 media professionals.
The results of the research are interesting. They confirm the results of previous research, which shows that journalists perceive themselves as an autonomous group that works for organizations that share common goals. This institutional vision, above the individual, is interesting for the definition of the journalistic profession. The summary could perhaps be made a little more precise. It is stated in a more generic way, when the basis of the work seems to be the way in which convergence has expanded the responsibilities of journalists to encompass tasks previously handled by other professionals within the newsroom. The occasions on which it is referring to all media workers, only journalists or only the rest of the staff should be specified a little better.
The methodology is well explained and details the sample used. Perhaps the way in which the 108 journalists/media professionals to whom the survey was initially sent could be clarified.
Regarding the differences in results between men and women, it would be interesting to place them in the broader context of the gender gap in the journalistic profession, and the way in which women differentially perceive their work in newsrooms. Issues such as less presence in positions of responsibility, greater precariousness or earlier abandonment of the profession can significantly influence these perceptions.
In the results section, the presentation of the numerical results has a lot of weight in the writing. Perhaps it would be interesting to include a summary table of the data (by categories, age, position, etc.) and reserve some more space in the text for the interpretation of this data and the comparison of the results between the different variables.
In the discussion, it would be interesting to also compare the results obtained with those of other recent similar surveys/studies/research in other European countries, to better contextualize the reality of journalistic practice in Portugal.
In summary, it is an interesting, well-conceived study, with interesting results.
Author Response
Dear Reviewer,
Thanks for your thorough review of the paper and your valuable feedback. All suggestions have been incorporated, highlighted in yellow. Key changes include:
- We briefly introduced the concept of the convergence culture, expanding journalists' roles to include tasks previously handled by other professionals.
- The text underwent multiple revisions, clarifying the distinctions between journalists, photojournalists, and techno-actors.
- Email addresses were gathered from media outlet websites or provided by editors in response to the authors' inquiries.
- Throughout the text, we offer reflections on the impact of gender differences in newsrooms.
- We added two tables presenting key data from the article, enhancing readability of the results.

Reviewer 2 Report
Comments and Suggestions for Authors
Dear authors,
Thank you for the opportunity to read the submitted article and your contributions.
I believe that the text provides important knowledge by being based on an empirical study that analyzes the work of social communication professionals based on their self-representations. Even though it is focused on a single country, dealing with the Portuguese context, I attribute relevance precisely because it allows the study to be replicable in other contexts.
The text is well structured, with internal coherence and written clearly and fluently.
The literature review that supports the theoretical-conceptual basis is adequate and carried out properly. It handles referential sources for the approach taken, although it seems to us that in some passages aspects such as the influence of non-human actants on the execution of work in newsrooms due to the highly digitalized context and the very mention of what connects with the flows of work based on what established convergence. In this sense, we recommend adding what authors referenced in the text (such as Zamith, Rodrigo, and Oscar Westlund. 2022) bring about non-human actants.
We also indicate a reference that can greatly contribute to expanding the perspective worked in the text based on the findings of the empirical study, as it works on the concept of post-convergence. This conception seems to help understand the professional contexts in journalistic newsrooms strongly impacted by digitalization. See: Ramírez, R. (2020). Post-Convergent Mediatization: toward a media typology beyond web 2.0. Mediatization Studies, v. 4, 9-23. DOI: 10.17951/ms.2020.4.9-23
We also left the following questions for the authors, as we missed them being mentioned in the methodology section:
- Since the form was applied in 2022, why did the study not include information about the impact of the Covid-19 pandemic on the workflows of the newsrooms studied?
- What factors were observed in the Portuguese media studied that were contributed by working remotely (home office) during the Covid-19 pandemic?
- What aspects were observed by professionals in relation to hybrid work, or was this form of work not identified in Portuguese environments during the period studied?
Thank you once again for the opportunity to read and evaluate the article.
Author Response
Dear Reviewer,
Thanks for your thorough review of the paper and your valuable feedback. We have introduced or highlighted suggested references (see text in yellow). While addressing the impact of Covid-19 on journalistic practices an important topic, it was not the focus of our study. Additionally, there are no questions about the pandemic in the online survey we administered. Therefore, it is impossible to answer some of the questions you kindly raised in your review. We believe that this impact would be better analyzed in the Portuguese context with a dedicated survey focused on the subject.

Reviewer 3 Report
Comments and Suggestions for Authors
The research requires a general rethinking in order to focus specifically on the new roles that are demanded in the journalistic market in newsrooms, the profiles that are detected and how working relationships are articulated within the framework of news production. Thus, both the title and the objectives of the research are too general and ambiguous. From the survey-based methodology we cannot analyse variables such as news quality or success, but rather journalists' perceptions of their work. There are interesting studies, such as the Annual Study on the Journalistic Profession (Madrid Press Association) that can provide interesting data. As the sample of the study and the conclusions focus on Portugal, this key word should also be included in the title.
Thus, the title and objectives of the research should focus on the profiles and roles of newsrooms in Portugal in the era of digital journalism. And in the objectives specify that it is a study that analyses the different profiles and their interrelationships, how collaboration is articulated, what productive routines it affects, as well as the roles that are demanded. The idea of success should be avoided or, in any case, redefined as a perception of success, as success itself can only be measured with an audience study.
By reordering the objectives and re-framing the research, the wording of the development would have to be reordered. The conclusions already point precisely to this new approach.
Author Response
Dear Reviewer,
Thank for your thorough review of the paper and for providing valuable feedback.
The paper has undergone an in-depth review and is now more focused on delineating the new roles required in today's journalistic market and how these roles manifest in work relationships within the realm of news production. The suggestions have been incorporated and are highlighted in yellow.
Key changes include:
- The suggested references have been integrated.
- The text has undergone multiple revisions to clarify the distinctions between journalists, photojournalists, and techno-actors.
- Two tables have been added to present key data from the paper, thereby enhancing the readability of the results.
- The word " Portuguese" has been included in the title. This makes perfect sense and is also in the keywords: "Portugal".
- It has been clarified that the "quality of news content" refers to the self-representation or perception of its creators (journalists, techno-actors, and photojournalists).
- The idea of "success" has been replaced with that of "professional satisfaction".
Thanks once again for your insightful comments and suggestions.
Reviewer 4 Report
Comments and Suggestions for Authors
I would point out the methodology: It is said that an online survey was used, but no information is given about the questions and the way of processing the data. Also, there is a hint to the correlation between media professionals and their capacity for interdependence in newsrooms, but no results of correlation analysis are provided.
Some definitions (e.g., what is in this study meant by "hybrid professionals") should be added. RQ1 does not need research, logical thinking and experience will take to the "yes" answer. At least, the question needs rephrasing to be more relevant.
Author Response
Dear Reviewer,
Thank for your thorough review of the paper and for providing valuable feedback.
The paper has undergone an in-depth review and is now more focused on delineating the new roles required in today's journalistic market and how these roles manifest in work relationships within the realm of news production. The suggestions have been incorporated and are highlighted in yellow.
Key changes include:
- Further details regarding the online survey questions were incorporated, along with enhancements in the Data and Methods section.
- The term "correlation" was substituted with "relationship" or "connection," aligning with the intended meaning of the authors. No statistical correlation test was conducted due to the non-probabilistic nature of the sample.
- The definition of hybrid professionals within the study's scope is now more explicit.
- RQ1 has been rephrased slightly, focusing on personal satisfaction, which was queried in the survey.

Round 2
Reviewer 3 Report
Comments and Suggestions for Authors
The authors have incorporated the suggestions made above and the quality of the text has improved significantly.